# Association of Long-Term Treatment by Botulinum Neurotoxins and Occupational Therapy with Subjective Physical Status in Patients with Post-Stroke Hemiplegia

**DOI:** 10.3390/toxins11080453

**Published:** 2019-08-02

**Authors:** Toyohiro Hamaguchi, Masahiro Abo, Kai Murata, Mari Kenmoku, Izumi Yoshizawa, Atsushi Ishikawa, Makoto Suzuki, Naoki Nakaya, Kensuke Taguchi

**Affiliations:** 1Department of Rehabilitation Medicine, The Jikei University School of Medicine, Tokyo 105-8461, Japan; 2Department of Rehabilitation, Graduate School of Health Sciences, Saitama Prefectural University, Saitama 343-8540, Japan; 3Department of Rehabilitation Medicine, The Jikei University School of Medicine Hospital, Tokyo 105-8461, Japan; 4Faculty of Health Sciences, Tokyo Kasei University, Saitama 350-1398, Japan

**Keywords:** BoNT-A, motor function, post-stroke hemiplegia, subjective physical status, upper extremity

## Abstract

The short-term effects of botulinum toxin type A (BoNT-A) treatment in stroke patients with upper extremity are well established. This study examined the association between the recovery of motor function of the upper extremity with subjective physical symptoms in outpatients receiving long-term BoNT-A and occupational therapy following stroke. We also investigated the expectations of patients who elected to continue treatment. Forty-seven patients (23 men and 24 women) aged 61 years received BoNT-A treatment more than 20 times. The subjective physical status was analyzed by using the visual analogue scale score through an eight-item questionnaire. Recovery of motor function in the upper extremity was detected by calculating the change (delta) in Fugl–Mayer Assessment (FMA), and ordinal logistic modeling analysis was used to determine the association between the delta-FMA score and the subjective level of agreement for each item. When the ordinal logistic modeling fit was statistically significant, results were interpreted as having logistic probability. The logistic curves discriminating one point (strongly disagree) from five points (strongly agree) were fit in a stepwise fashion. This study suggests that patients receiving long-term BoNT-A treatment and occupational therapy experienced an increased upper extremity mitigation and decreased insomnia after injection, regardless of the recovery of motor function.

## 1. Introduction

Patients with post-stroke hemiplegia often suffer from spasticity [1,2]. Severe spasticity combined with motor paralysis decreases the ability to perform activities typical of daily living [3,4]. Botulinum toxin type A (BoNT-A) inhibits the signal transmission by acetylcholine at the neuromuscular junction of the motor nerves, resulting in a muscle relaxation, and is effective in suppressing pain and spasticity [5,6]. BoNT-A treatment was insured in 2010 and has been used to treat spasticity in hemiplegic patients following stroke. BoNT-A treatment is recommended as a grade A treatment for stroke patients in the clinical practice guidelines in Japan, and is a strongly recommended treatment for severe hemiplegia [7].

The efficacy of BoNT-A is attenuated in three to four months following administration [8,9]. Therefore, suppressing the development of persistent spasticity and pain may contribute to the long-term maintenance and improvement of motor function [6,10]. Injections are repeated approximately every three months to maintain the efficacy of BoNT-A. The recovery of motor paralysis of the upper extremity may take more than six months post-stroke patients, indicating that doses of BoNT-A would be needed for a longer period [11,12]. These effects were verified by research of repeated BoNT-A administration for up to one and half years [9,10,11,12]. However, the effects of continuing BoNT-A treatment for several years remain unknown.

To effectively achieve rehabilitation in hemiplegic patients, it is necessary to understand the factors related to the motor skills of patients [13]. In particular, since severe symptoms of motor paralysis persist for long periods, it is beneficial for patients to be aware of their own prognosis and have realistic expectations of treatment effects before starting BoNT-A treatment. Physicians can provide patients with information about the effects of BoNT-A on muscle relaxation, reducing pain, and paralysis recovery [5,6].

In this study, we aimed to determine the association between the recovery in motor function of the upper extremity and muscle relaxation after treatment with subjective symptoms of pain in long-term outpatient treatment of BoNT-A and aid of rehabilitation after stroke. We also investigated the expectations for subjective physical status of patients who continued for treatments.

## 2. Results

Baseline characteristics of the 47 selected patients are shown in Table 1. The detailed diagnoses included 15 cases of cerebral infarction and 32 cases of cerebral hemorrhage, and the average number of months after onset was 56 (26–85) months (Table 1). Fugl–Mayer assessment (FMA) scores and responses to the subjective physical questionnaire (Table 2) are summarized in Table 3.

As a result of the subjective physical survey, the patients indicated that they felt both injection and exercise therapy were beneficial (Q6, mean visual analogue scale; VAS = 9.7 cm, Figure 1). In addition, the patients felt encouraged to continue treatment by doctors and therapists (Q9, mean VAS = 10.0). Patients psychologically accepted the sequelae of stroke (Q10, mean VAS = 10.0) and did not indicate any side effects of BoNT-A treatment (Q5, mean VAS = 0.0).

Results of the ordinal logistic modeling for the delta-FMA score and the subjective level of agreement for each questionnaire item are shown in Figure 2 and Figure 3. For comfortable sleep (Q2) and mitigation (Q3), the ordinal logistic modeling fit was a statistically significant, valid logistic probability. The delta-FMA score and the subjective level of agreement for attenuation of pain (Q1) showed a tendency toward significance in logistic modeling (*p* = 0.070). The logistic curves discriminating one point (strongly disagree with the question) from five points (strongly agree with the question) showed a stepwise fashion fit (item 2, *p* = 0.02; item 3, *p* = 0.04; Table 4).

Items from the questionnaire given to patients before BoNT-A treatment that showed a statistically significant fit in ordinal logistic modeling and VAS are shown in Table 5.

## 3. Discussion

This study aimed to verify the association between the recovery of motor function of the upper extremity and subjective symptoms, such as muscle relaxation and pain after treatment, in patients with continued BoNT-A treatment and upper extremity exercise therapy for more than five years after stroke. Our results suggested that long-term treatment could improve subjective symptoms of sleeplessness and arm weakness caused by sequelae of stroke without recovery of motor function in upper extremity, while also decreasing pain. BoNT-A treatment for sequelae after stroke has shown short-term therapeutic effects within 12 months [12,14,15,16]. Additionally, VAS is commonly used for pain assessment [17,18]. This study is the first survey, to our knowledge, to show the effects of continuous treatment for over five years.

When providing long-term treatment, it is important to explain the expected benefits and potential side effects to patients [19,20,21]. Although BoNT-A treatment has been shown to have negative side effects [19,22], patients in this study did not report any. It was suggested that patients were conscious that the treatment had less physical burden. By interviewing patients before starting BoNT-A treatment, clinicians can explain potential benefits unrelated to regaining motor function in cases of severe paralysis with symptoms of sleep disturbance and upper limb discomfort.

BoNT-A is a recommended first-line treatment for post-stroke spasticity affecting the upper extremities in adults [23,24]. The results of this study suggested that long-term BoNT-A treatment reduced these negative physical symptoms. The long-term effects of this combined therapy, including suppressed spasticity, improved insomnia consequent to increased muscle tone, and the mitigation of sensations in the arm, may have been dependent on underlying mechanisms.

According to subjective survey results, the responses of the doctor and therapist motivated patients to continue long-term BoNT-A treatment. In addition, it was suggested that the patients accepted the sequelae associated with stroke with hopes for further treatment. BoNT-A treatment is known not only to relieve physical symptoms, but also to maintain physical function when combined with rehabilitation [4,25]. Physical and/or psychological support is also important to patients continuing rehabilitation and long-term treatment.

These results may be useful as a supplementary resource for physicians who recommend BoNT-A treatment for hemiplegic stroke patients. Physicians can use similar questions to assess patient symptoms and provide patient-specific recommendations and prognoses. Our results also indicate that patients can actively perform rehabilitation without suffering from sequelae, and that the subjective physical status can predict with 60% accuracy that patients will suffer less sequelae even if their upper extremity motor function does not improve after prolonged treatment with BoNT-A.

There are several limitations associated with this study. Firstly, the subjective physical survey items have not been validated. The use of a subjective evaluation limits the reproducibility of this study and makes it difficult to translate the findings to larger populations. Subjective physical characteristics, such as the mitigation of pain, sleeplessness, and arm sensations should be investigated using an objective rating scale to improve translatability. Therefore, the association between motor function and subjective physical symptoms is considered to be underestimated in this study. BoNT-A treatment is intended to improve physical symptoms, and more convincing explanations for patients should use objective indicators. First, further study should use objective measurements of physical symptoms (pain, sleeplessness, and arm mitigation) to verify our findings. Second, selection bias could be limiting the effects of this study because the patients included were only those who received BoNT-A treatment more than 20 times. Therefore, it is likely that those who had significantly recovered upper extremity motor function during the course of treatment and those who stopped treatment due to deterioration in physical condition were not included in this survey. The magnitude of this bias is unknown. It is necessary to investigate whether similar results can be obtained for patients with a short period of treatment. Third, the study was likely underpowered based on the sample size calculations that were performed, which could contribute to increased type 2 errors. It is also difficult to predict if similar results can be obtained. To fully and scientifically examine the physical effects of BoNT-A treatment, it will also be necessary to recruit and reanalyze patients who comply with long-term monitoring, as this would reduce type 2 errors in future studies. Finally, the data we obtained from patients were limited and confounding variables may have been present, these include the use of sleeping pills, analgesics, or other medications taken for symptom management. Given these limitations, the apparent next step would involve the use of study data to explain BoNT-A treatment to patients and thus allow them to feel safe and receive continuous treatment. However, patients receiving this treatment should be monitored, and improvements in treatment compliance should be verified.

## 4. Conclusions

This study suggests that patients who received BoNT-A treatment more than 20 times and continued occupational therapy for at least five years experienced increased upper extremity mitigation and decreased insomnia after BoNT-A injection, even without the recovery of motor function.

## 5. Materials and Methods

This was a retrospective, longitudinal study. Patients underwent BoNT-A treatment at least 20 times at Tokyo Jikei University Hospital from November 2010 to March 2018. The written approval of consent to receive BoNT-A treatment was obtained from all 47 patients enrolled in the study. This study was approved by the ethics committee of the Tokyo Jikei University School of Medicine (No. 24-295-7061, 4 February 2013). The registry was carried out according to the principles of the declaration of Helsinki.

Motor function of the upper extremity in patients with post-stroke hemiplegia and their subjective physical status were examined after BoNT-A treatment. Assessment of motor function of the upper extremity was performed using the Fugl–Mayer assessment (FMA) score [26]. In addition, a subjective physical survey was conducted regarding the combination of BoNT-A treatment and occupational therapy. The dosage, injection site, and FMA data were obtained retrospectively from patient medical records, and the last FMA evaluation and subjective physical survey were conducted by the occupational therapist responsible for the patients (Figure 4).

### 5.1. Subjects

The inclusion criteria were (1) patients with post-stroke hemiplegia who were continuing BoNT-A treatment and occupational therapy, (2) patients who consented to BoNT-A treatment, and (3) patients who had at least 20 BoNT-A injections (Figure 5). Patients with moderate to severe spasticity following stroke were considered for this study.

A sample size of 53 was derived by logistic regression z-tests insertion of power (0.80), α (0.05), and *R^2^* (0.60) values in G*Power 3.1 software [27], 47 patients who fit the inclusion criteria were enrolled in this study, clinical characteristics was shown in Table 1.

### 5.2. BoNT-A Injections

Patients received injections of BoNT-A OnabotulinumtoxinA (0.90 ng/100 U vial Botox®, Allergan, Irvine, CA, USA). One vial of BoNT-A (100 U) was diluted with 2 mL saline. Selection of muscles and injected doses were determined by the physician based on clinical assessment. Treated upper extremities were injected with BoNT-A (0–60 U) into the selected muscles at up to four sites (Table 6).

### 5.3. Functional Assessment of Upper Extremity

FMA is a comprehensive evaluation index for assessing motor function of hemiplegic patients after stroke, including detecting sensorimotor impairments [26]. The FMA score has been used to verify evaluations in physiotherapy and BoNT-A therapy, specifically for the upper extremity [25].

### 5.4. Items of the Subjective Physical Questionnaire

Subjective physical survey items were created by physicians, occupational therapists, and clinical psychologists (Table 2). Answers were automatically recorded on a 10 cm visual analogue scale (VAS), which marked positions where patients possessed feeling. The length of the measured VAS was converted to a score, (1) 0–2 cm; (2) 2–4 cm; (3) 4–6 cm; (4) 6–8 cm; or (5) 8–10 cm.

### 5.5. Data Analysis

To analyze recovery of motor function in the upper extremity, delta-FMA was calculated from the initial and last FMA scores obtained during BoNT-A treatment.

The association between the delta-FMA score and each item from the subjective physical status questionnaire was determined by ordinal logistic modeling analysis [28]. The principle of ordinal logistic modeling is to fit the probability (*p*) of multiple dichotomous responses (Equation (1)) to a linear model (Equation (2)):(1)$$g\left(x\right)=\frac{1}{1+e^{-f\left(x\right)}}$$
(2)$$\;f\left(x\right)=\beta_{0}+\beta_{1}x+e\;$$
where *x* is the explanatory variable, *β_i_* is the partial regression coefficient, and *e* is the residual between actual and predicted data. Therefore, for multilevel ordinal responses, the cumulative probability is calculated at each level to generate a simple regression. In this study, the probability of five levels of subjective agreement was evaluated in association with delta-FMA score. All statistical analyses were performed using the R 3.5.2 software (R Foundation for Statistical Computing, Vienna, Austria).

## Figures and Tables

**Figure 1 toxins-11-00453-f001:**
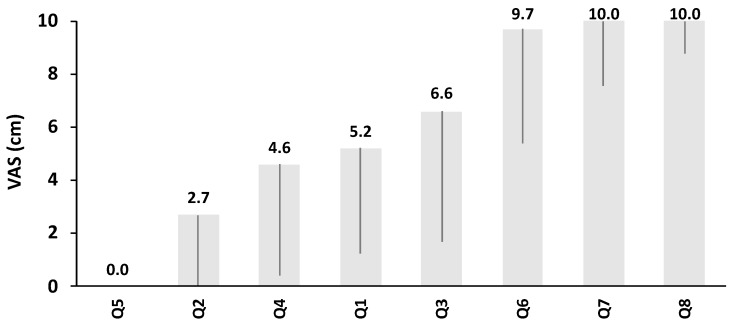
Subjective physical survey results. The X-axis shows items of the subjective physical survey. Results of visual analogue scale (VAS) are shown in the Y-axis. Bars indicate the mean and standard deviation of VAS (cm).

**Figure 2 toxins-11-00453-f002:**
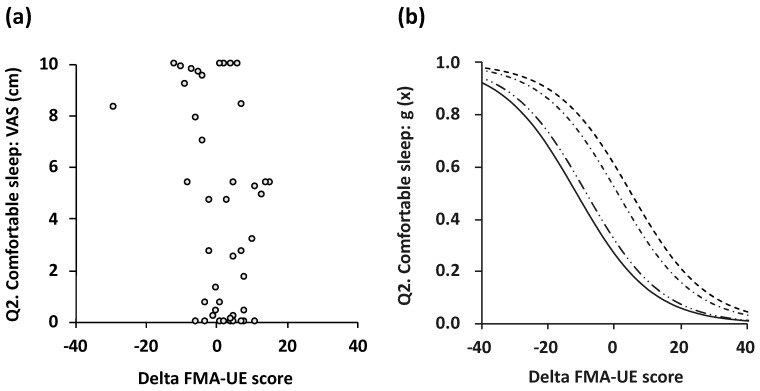
Scatterplots (**a**) and logistic probability plots (**b**) showing the association between subjective level of agreement for item 2 and delta FMA score. The logistic curve that discriminated 1 point (strongly disagree with the question) from 2 points (disagree with the question; dashed line), 2 points from 3 points (neither agree nor disagree; dot-dash line), 3 points from 4 points (agree; two-dot chain line), and 4 points from 5 points (strongly agree; solid line) was in a stepwise fashion fit (*p* = 0.002). FMA: Fugl–Meyer assessment.

**Figure 3 toxins-11-00453-f003:**
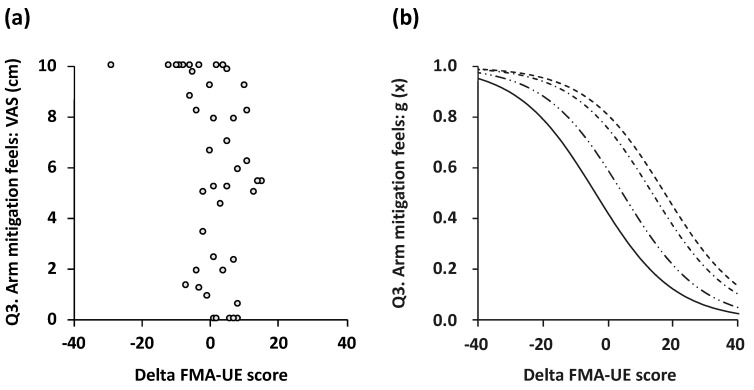
Scatterplots (**a**) and logistic probability plots (**b**) showing the association between subjective level of agreement for item 3 and delta FMA score. The logistic curve that discriminated 1 point (strongly disagree with the question) from 2 points (disagree with the question; dashed line), 2 points from 3 points (neither agree nor disagree; dot-dash line), 3 points from 4 points (agree; two-dot chain line), and 4 points from 5 points (strongly agree; solid line) was in a stepwise fashion fit (*p* = 0.004). FMA: Fugl–Meyer assessment.

**Figure 4 toxins-11-00453-f004:**
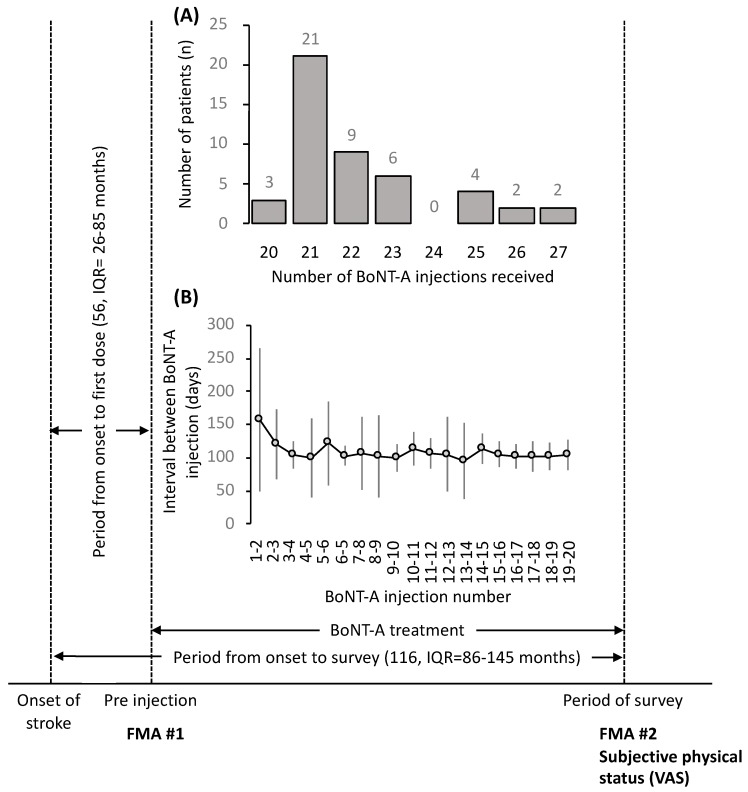
Study design and result of Botulinum toxin type A (BoNT-A) treatment. The X-axis shows the period from stroke onset to the start of BoNT-A treatment, the period from onset to subjective physical investigation, and the period of BoNT-A treatment. FMA was performed pre- and post-BoNT-A treatment, and the subjective physical survey was performed after the final BoNT-A injection. (**A**) Frequency distribution (20–27) of patients within the BoNT-A treatment period. (**B**) The number of days between intervals of BoNT-A treatment is shown over the injection course.

**Figure 5 toxins-11-00453-f005:**
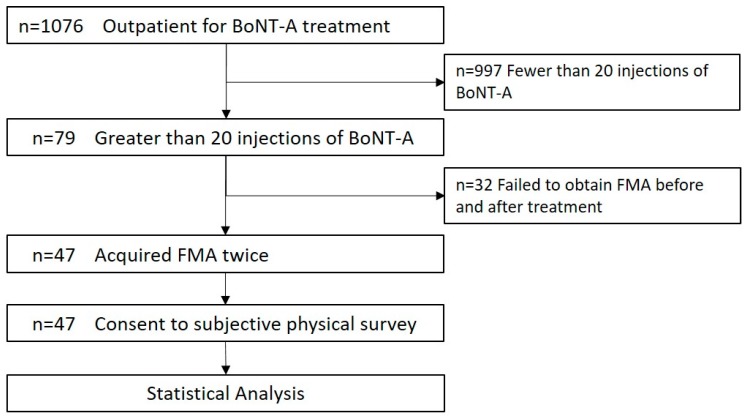
Patient selection and inclusion criteria. Procedure of data acquisition and selection for analysis.

**Table 1 toxins-11-00453-t001:** Characteristics of patients at survey.

**Pre-Treatment**	**Value**
Sex (female:male)	23:24
Type of Stroke (%)	
infarction	15 (32)
hemorrhage	32 (68)
Period from onset to first dose (months)	56 (26–85)
**Post-Treatment**	**Value**
Age	61 ± 16
Period since onset (months)	116 (86–145)
Paralysis side (Right:Left)	21:26
Number of BoNT-A injections (median (min-max))	21 (20–27)
BoNT-A dose (Units/patient)	268 ± 77
Interval between doses (days)	108 ± 39

Values are mean ± standard deviation (SD), n, or median (interquartile range).

**Table 2 toxins-11-00453-t002:** Questions for patients continuing BoNT-A treatment and occupational therapy following stroke.

**How do you feel the efficacy and side effects of BoNT-A treatment given for sequelae of stroke as subjective symptoms?**
Q1. Pain in my limbs is relieved when I receive an injection.
Q2. I will sleep better when injected.
Q3. My arm feels mitigation when I am injected.
Q4. Injections are painful for me.
Q5. I feel that the injection has side effects.
**Do you expect BoNT-A treatment and occupational therapy to be a motivation to continue treatment?** **Furthermore, is the response of doctors and therapists encouraging for you?**
Q6. I think it is important to me to continue injection and rehab.
Q7. I am encouraged by the response of doctors and therapists.
Q8. I think it is important to accept sequelae of stroke.

**Table 3 toxins-11-00453-t003:** Classification of psychological status after long term BoNT-A treatment for patients with paralysis of the upper extremity.

	1: 0–<2 cm	*n*	2: 2–<4 cm	*n*	3: 4–<6 cm	*n*	4: 6–<8 cm	*n*	5: 8–10 cm	*n*
Q1	32 (27–37)	18	30 (28–42)	3	41 (39–43)	5	30 (28–39)	4	26 (21–32)	17
Q2	30 (23–37)	21	37 (33–41)	4	40 (37–46)	8	28 (28–29)	2	26 (24–28)	12
Q3	30 (26–28)	11	28 (27–37)	3	40 (37–42)	8	32 (30–39)	7	27 (24–32)	18
Q4	29 (26–34)	21	23 (20–26)	2	32 (26–42)	7	39 (39–40)	2	32 (26–38)	15
Q5	29 (25–38)	41	36 (34–37)	2	34 (33–36)	2	-	0	30 (29–31)	2
Q6	43	1	-	0	41	1	24	1	31 (26–38)	44
Q7	-	0	-	0	39 (34–47)	3	30 (24–32)	5	32 (26–38)	39
Q8	32 (32–43)	3	-	0	32 (24–38)	5	41 (39–45)	4	28 (25–36)	35

FMA score and number of patients were represented based on visual analogue scale. Scores are FMA-upper extremity (UE) median, interquartile range (IQR, 25–75%) and number of patients (*n*). “1: 0–<2” indicates class 1 if VAS is 0 to less than 2 cm.

**Table 4 toxins-11-00453-t004:** Delta Fugl-Meyer Assessment-Upper Extremity (FMA-UE) estimated visual analogue scale (VAS) scores by a logistic ordinal regression model.

Question Item	β	SEM	Z	*p*
Q1	−0.07	0.04	−1.81	0.07
Q2	−0.09	0.04	−2.27	0.02
Q3	−0.08	0.04	−2.07	0.04
Q4	0.00	0.04	0.01	0.99
Q5	0.04	0.06	0.66	0.51
Q6	−0.07	0.09	−0.74	0.46
Q7	−0.06	0.06	−1.03	0.30
Q8	−0.08	0.05	−1.54	0.12

**Table 5 toxins-11-00453-t005:** Clinical questions for BoNT-A selection.

**How do you feel your symptoms due to stroke’s sequelae?**
Q1. Are you suffering from pain in your arm?
Q2. Do the symptoms of your arm interfere with your sleep?
Q3. Do you feel that your arms are dull?
**What are your thoughts about BoNT-A treatment and occupational therapy?**
Q4. Do you think it is important to you to continue rehabilitation?
Q5. Do you think that doctors and therapists will be encouraging for you?
Q6. Do you think it is important to accept your stroke’s sequelae?

**Table 6 toxins-11-00453-t006:** Injected BoNT-A dosage into specific muscle.

Clinical Anatomical Position	Treatment Muscle	Dosage (Units/Injection)
Adducted upper arm	*Pectoralis major*	34.3 ± 14.7
	*Latissimus dorsi*	21.3 ± 16.8
	*Teres major*	2.5 ± 6.3
Extended elbow	*Triceps brachii*	2.4 ± 7.0
Flexed elbow	*Biceps brachii*	37.4 ± 16.9
	*Brachialis*	3.1 ± 7.1
	*Brachioradialis*	1.0 ± 2.6
Pronated forearm	*Pronator teres*	12.0 ± 13.0
	*Pronator quadratus*	0.3 ± 1.1
Flexed wrist	*Flexor carpi radialis*	29.3 ± 16.5
	*Flexor carpi ulnaris*	7.1 ± 8.5
	*Palmaris Iongus*	0.2 ± 0.6
Clenched fist	*Flexor digitorum superficialis*	33.9 ± 16.4
	*Flexor digitorum profundus*	6.0 ± 9.0
Thumb-in-palm	*Adductor pollicis*	14.0 ± 11.7
	*Flexor pollicis longus*	4.6 ± 6.8
	*Flexor pollicis brevis*	2.1 ± 5.3
	*Lumbricals*	1.3 ± 3.8

Values are mean ± standard deviation. BoNT-A; botulinum neurotoxin type-A.

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
