# Peer review of "Association of Long-Term Treatment by Botulinum Neurotoxins and Occupational Therapy with Subjective Physical Status in Patients with Post-Stroke Hemiplegia"

_toxins, 2019, doi:10.3390/toxins11080453_

Round 1
Reviewer 1 Report
This study has dealt with long-term treatment by botulinum neurotoxins combined with occupational therapy in patients with post stroke hemiplegia.
Study results could verify the association between recovery of motor function of the upper extremity and subjective symptoms, such as muscle relaxation and pain after treatment in patients with continued BoNT-A treatment and upper extremity exercise therapy for more than 5 years after stroke. This is the first study looking at the long-term effect and hence is novel. This is also the first study looking at overall pain on VAS for long term in these patients receiving the combined therapy. Authors did not identify any side effects.
In terms of safety assessments, did the author used any questionnaire or metrics or was it just a subjective report? Please identify.
There is no discussion on potential mechanisms underlying this long-term effect of the combined therapy. Authors are encouraged to add a section in the discussion for this missing point.
Considering limitations of the study, also highlighted by the authors, what will be the next step, or clinical guideline? Authors are encouraged to add a few lines at the end of their manuscript to indicate whether conclusion from this study can be applied or further investigation is required. Authors are also encouraged to take a note about the internal and external validity of their data.
Author Response
Response to Reviewer 1 Comments
Reviewer #1:
Study results could verify the association between recovery of motor function of the upper extremity and subjective symptoms, such as muscle relaxation and pain after treatment in patients with continued BoNT-A treatment and upper extremity exercise therapy for more than 5 years after stroke. This is the first study looking at the long-term effect and hence is novel. This is also the first study looking at overall pain on VAS for long term in these patients receiving the combined therapy. Authors did not identify any side effects.
Comment 1:
In terms of safety assessments, did the author used any questionnaire or metrics or was it just a subjective report? Please identify.
We thank Reviewer #1 for reviewing our manuscript and for the helpful comments. We have corrected the indicated points.
Answer to comment 1:
We agree with this comment and have added the following text to the Discussion section:
“Subjective physical characteristics such as the mitigation of pain, sleeplessness, and arm sensations should be investigated using an objective rating scale to improve translatability. Furthermore, the subjective physical survey items have not been validated and, therefore, the internal and external validity could not be verified. A future study should use quantitative measurements of physical symptoms to validate and verify these results.”
Please see page 5, lines 137–142 in the revised manuscript.
Comment 2:
There is no discussion on potential mechanisms underlying this long-term effect of the combined therapy. Authors are encouraged to add a section in the discussion for this missing point.
Answer to comment 2:
We agree with this comment and have added the following sentences to the Discussion section:
“BoNT-A is a recommended first-line treatment for post-stroke spasticity affecting the upper extremities in adults [23,24]. The results of this study suggest that long-term BoNT-A treatment reduced these negative physical symptoms. The long-term effects of this combined therapy, including suppressed spasticity, improved insomnia consequent to increased muscle tone, and mitigation of sensations in the arm, may depend on underlying mechanisms.”
Please see page 5, lines 119–123 in the revised manuscript.
Comment 3:
Considering limitations of the study, also highlighted by the authors, what will be the next step, or clinical guideline? Authors are encouraged to add a few lines at the end of their manuscript to indicate whether conclusion from this study can be applied or further investigation is required. Authors are also encouraged to take a note about the internal and external validity of their data.
Answer to comment 3:
We have added the following text to the Discussion section to address this comment:
“Given these limitations, the apparent next step would involve using the study data to explain BoNT-A treatment to patients and thus allow them to feel safe and receive continuous treatment. However, patients receiving this treatment should be monitored, and improvements in treatment compliance should be verified. To fully and scientifically examine the physical effects of BoNT-A treatment, it will also be necessary to recruit and reanalyze patients who cooperate with long-term monitoring, as this would reduce type 2 errors in future studies.”
Please see page 5, lines 151–156 in the revised manuscript.
Reviewer 2 Report
The article entitled "Association of long-term treatment by botulinum neurotoxins and occupational therapy with subjective physical status in patients with post-stroke hemiplegia" is short survey study with many confounds associated. Authors have three important limitations 1) the reproducibility of this study and makes it difficult to translate to larger populations 2) selection bias could be limiting the effects of this study because the patients included were only those who received BoNT-A treatment more than 20 times and 3)under powered based on the sample size calculations that were performed, which could contribute to increased type 2 errors.
Also involvement of other factors such as sleeping pills, analgesics, or other medications taken for symptom management is not addressed.
Furthermore, these results also indicate that patients can actively perform rehabilitation without suffering from sequelae, and that subjective physical status can predict that patients will suffer less sequelae even if their upper extremity motor function does not improve after prolonged treatment with BoNT-A with 60% accuracy.
Authors should address these issue elaborately in the discussion under limitations and future directions.
Author Response
Response to Reviewer 2 Comments
Reviewer #2:
The article entitled "Association of long-term treatment by botulinum neurotoxins and occupational therapy with subjective physical status in patients with post-stroke hemiplegia" is short survey study with many confounds associated. Authors have three important limitations 1) the reproducibility of this study and makes it difficult to translate to larger populations 2) selection bias could be limiting the effects of this study because the patients included were only those who received BoNT-A treatment more than 20 times and 3)under powered based on the sample size calculations that were performed, which could contribute to increased type 2 errors.
Also involvement of other factors such as sleeping pills, analgesics, or other medications taken for symptom management is not addressed.
Furthermore, these results also indicate that patients can actively perform rehabilitation without suffering from sequelae, and that subjective physical status can predict that patients will suffer less sequelae even if their upper extremity motor function does not improve after prolonged treatment with BoNT-A with 60% accuracy.
Comment 1:
Authors should address these issue elaborately in the discussion under limitations and future directions.
Answer to comment 1: We thank Reviewer #2 for reviewing our manuscript and for these helpful comments. We have added the following text to the Discussion section to address this comment:
“Given these limitations, the apparent next step would involve using the study data to explain BoNT-A treatment to patients and thus allow them to feel safe and receive continuous treatment. However, patients receiving this treatment should be monitored, and improvements in treatment compliance should be verified. To fully and scientifically examine the physical effects of BoNT-A treatment, it will also be necessary to recruit and reanalyze patients who cooperate with long-term monitoring, as this would reduce type 2 errors in future studies.”
Please see page 5, lines 151–156 in the revised manuscript.
Round 2
Reviewer 2 Report
Authors try to address the questions raised by the reviewers. Since there are several limitations in the study validity of the study might be compromised. Concerning this authors should able to address the how they are going to handle these limitations in most approachable manner. I would appreciate if you could make easy for the readers about this.
Author Response
Response to Reviewer 2’s Comment
Reviewer #2:
Authors try to address the questions raised by the reviewers. Since there are several limitations in the study validity of the study might be compromised. Concerning this authors should able to address the how they are going to handle these limitations in most approachable manner. I would appreciate if you could make easy for the readers about this.
Comment 1
Authors should address these issue elaborately in the discussion under limitations and future directions.
Response: We thank reviewer #2 for reviewing our revised manuscript and providing helpful comments. We have rewritten the following text in the Discussion section to address your concern about our study limitations (page 5, line 136 to page 6, line 159):
“There are several limitations associated with this study. Firstly, the subjective physical survey items have not been validated. The use of a subjective evaluation limits the reproducibility of this study and makes it difficult to translate the findings to larger populations. Subjective physical characteristics such as the mitigation of pain, sleeplessness, and arm sensations should be investigated using an objective rating scale to improve translatability. Therefore, the association between motor function and subjective physical symptoms was considered to be underestimated in this study. BoNT-A treatment is intended to improve physical symptoms, and more convincing explanations for patients should use objective indicators. Further study should use objective measurements of physical symptoms (pain, sleeplessness, and arm mitigation) to verify our findings. Secondly, selection bias could be limiting the effects of this study because the patients included were only those who received BoNT-A treatment more than 20 times. Therefore, it is likely that those who had significantly recovered upper extremity motor function during the course of treatment and those who stopped treatment due to deterioration in physical condition were not included in this survey. The magnitude of this bias is unknown. It is necessary to investigate whether similar results can be obtained for patients with a short period of treatment. Thirdly, the study was likely underpowered based on the sample size calculations that were performed, which could contribute to increased type 2 errors. It is also difficult to predict if similar results can be obtained. To fully and scientifically examine the physical effects of BoNT-A treatment, it will also be necessary to recruit and reanalyze patients who comply with long-term monitoring, as this would reduce type 2 errors in future studies. Finally, the data we obtained from patients were limited and confounding variables may have been present, including the use of sleeping pills, analgesics, or other medications taken for symptom management. Given these limitations, the apparent next step would involve the use of study data to explain BoNT-A treatment to patients and thus allow them to feel safe and receive continuous treatment. However, patients receiving this treatment should be monitored, and improvements in treatment compliance should be verified.”